# Influence of Long-Period Stacked Ordered Phases on Inductive Impedance of Mg-Gd-Y-Zn-Zr-Ag Alloys

**DOI:** 10.3390/ma16020640

**Published:** 2023-01-09

**Authors:** Shiyuan Xu, Chuming Liu, Yonghao Gao, Shunong Jiang, Yingchun Wan, Zhiyong Chen

**Affiliations:** 1School of Materials Science and Engineering, Central South University, Changsha 410083, China; 2Light Alloy Research Institute, Central South University, Changsha 410083, China

**Keywords:** Mg-Gd-Y-Zn-Zr-Ag, electrochemical impedance spectroscopy, LPSO phase, dual inductive loop

## Abstract

In this paper, the influence of long-period stacked ordered (LPSO) phases on the electrochemical impedance spectroscopy (EIS) of a Mg-Gd-Y-Zn-Zr-Ag alloy in 0.9 wt.% NaCl was investigated. The Mg-6Gd-3Y-1Zn-0.5Zr-0.3Ag (wt.%) alloy samples with and without LPSO phases in the grain interior (HOMO and LPSO, respectively) were prepared using different heat treatments. The EIS results showed that both the HOMO and LPSO samples’ Nyquist diagrams contained two inductive loops. However, in the Nyquist plots of the LPSO samples, the inductive loops at 1.71–0.67 Hz appeared in the first quadrant rather than the fourth quadrant. Analysis of the fitting parameters illustrated that the abnormal shape of the inductive loops is related to greater values of the surface film capacitance *C_f_* and double layer capacitance *C_dl_* in the LPSO samples. Further investigations through corrosion morphology observation indicated that the greater values of *C_f_* and *C_dl_* in the LPSO samples resulted from the existence of intragranular LPSO phases that created more film-free areas. The above results show that a better understanding of the relationship between the inductive impedance and corrosion morphology of a Mg-6Gd-3Y-1Zn-0.5Zr-0.3Ag alloy in 0.9 wt.% NaCl solution was attained.

## 1. Introduction

Due to their low density and high mechanical properties, Mg-RE-Zn alloys are remarkably promising for applications in the aerospace and transportation industries [1,2,3,4,5,6,7]. However, their shortage of corrosion properties has been the main barrier to their wider application [8,9,10,11]. Except for their high corrosion rates, susceptibility to localized corrosion is another critical problem, which also dramatically reduces the safety and service life of related products [12,13,14]. What is more, the damage of localized corrosion is hard to find due to the cover of corrosion products. Normally, the damage situation is assessed by observing the corrosion morphology after removing the corrosion products, which is impossible for the working components of Mg-RE-Zn alloys. In this case, it is very necessary to find a suitable technique to monitor and investigate localized corrosion.

Corrosion characteristics can be quantitatively studied using various techniques, i.e., weight loss tests, hydrogen evolution tests as well as electrochemical methods. While weight loss tests and hydrogen evolution tests can provide indispensable benefits associated with corrosion rate measurements, it is infeasible to investigate other information such as corrosion mechanisms and morphology. Electrochemical impedance spectroscopy (EIS) is an effective and nondestructive method for investigating the corrosion behavior of Mg alloys [15,16,17,18]. It helps to disclose the correlation between the surface morphological evolution and the corrosion process itself, which is critical for understanding the corrosion mechanisms. In aggressive solutions, impedance diagrams of Mg alloys normally contain one or two capacitance loops at high and medium frequencies and one inductive loop at low frequencies [15,16,17,18,19,20,21]. Further studies have indicated that the capacitance loops represent the charge transfer process across the capacitive double layer and the accumulated corrosion products, and the inductive loops represent the relaxation process of intermediate species (mainly ${\mathrm{Mg}}^{+}$) [22,23,24,25,26,27]. In this case, the corrosion behavior of Mg alloys can be characterized by the shape of EIS diagrams and the values of the fitting parameters.

Normally, EIS analysis of Mg corrosion was limited in the capacitance loops; inductive impedance was disregarded [17,18,19,27]. However, recent studies have illustrated that the inductive impedance data in EIS also help to understand the corrosion behavior of Mg alloys [28,29,30]. It is worth noting that the EIS data of some Mg alloy samples contain dual inductive loops (other than the normally recognized inductive loop) at low frequencies [31,32,33,34,35]. The study on Mg-Si alloys by Cao et.al [32] claimed that the appearance of dual inductive loops was related to the nonuniform corrosion morphology of Mg alloys, i.e., the two inductive loops represented the relaxation process of ${\mathrm{Mg}}^{+}$ ions in the lightly and severely corroded areas, respectively. These results indicated that the appearance of dual inductive loops could reflect the corrosion morphology, which helps us to monitor and investigate localized corrosion, while some researchers believed that the dual inductive loops represented the relaxation processes of different adsorbed species rather than nonuniform corrosion morphology [31,34]. These disagreements on the origins of the dual inductive loops make it impossible to remark any consistent conclusions on the corrosion mechanism.

The presence of long-period stacked ordered (LPSO) phases is the most astonishing characteristic of Mg-RE-Zn alloys and are responsible for their remarkable mechanical properties. However, such LPSO phases may dramatically affect the corrosion rate and how the alloy corrodes [36,37,38,39,40,41]. As mentioned above, differences in the corrosion process may affect the shape of EIS diagrams. However, the related studies were rather limited; therefore, further exploration is required.

In our previous work, the EIS data of the homogenized Mg-6Gd-3Y-1Zn-0.5Zr-0.3Ag (wt.%, VW64M) alloy samples were also observed to contain dual inductive loops when the samples suffered from severe localized corrosion and exhibited nonuniform corrosion morphology [42]. In addition, the VW64M Mg alloys also contained plenty of LPSO phases. The variation of the LPSO phases might have affected the corrosion morphology and corrosion type of the VW64M Mg alloys, and the shape of inductive impedance might have changed accordingly. The investigation in inductive impedance might help understand the influence of LPSO phases on the damage of localized corrosion in VW64M Mg alloys.

There are two main purposes for this paper: (1) to verify whether the appearance of dual inductive loops is associated with the nonuniform corrosion morphology caused by localized corrosion and (2) to verify the influence of intragranular LPSO phases on the inductive impedance and related mechanism. In this case, VW64M samples with and without intragranular LPSO phases were prepared through different heat treatments. The EIS data of both samples were tested. The difference in the inductive impedance of both samples was analyzed by characterizing the corrosion morphologies and quantitative fitting of the EIS diagrams.

## 2. Materials and Methods

The preparation method and chemical composition of experimental VW64M alloys were presented in our past research [42]. The as-cast ingot was cut into blocks with a volume of 10 mm × 10 mm × 5 mm. Samples without intragranular LPSO phases were fabricated by homogenization at 510 °C for 24 h and subsequent water-quenching at room temperature, which were hereby designated as the HOMO samples. The samples with intragranular LPSO were homogenized at 510 °C for 24 h and annealed at 480 °C for 2 h to precipitate LPSO phases in the grain interior, which were hereby termed as the LPSO samples.

Microstructure observation for the HOMO and LPSO samples was carried out by scanning electron microscopy (SEM, TESCAN MIRA2, Tescan, Brno, Czech Republic). The phase composition of the two samples was measured by X-ray diffraction (XRD, Rigaku D/Max 2500, Rigaku, Tokyo, Japan) using a copper target. The scan range of 2θ was from 10° to 80° with a scanning rate of 8°/min. The XRD pattern results were identified using MDI Jade 6.5 software.

EIS tests were carried out with an electrochemical workstation (PARSTAT PMC2000, Princeton, Oak Ridge, TN, USA). A standard three-electrode cell was used: The HOMO and LPSO samples served as the working electrode with an exposed surface of 1 cm^2^; the reference electrode was a saturated calomel electrode (SCE); the counter electrode was a platinum electrode. For chloride, ions were the aggressive ions that commonly exist in the surrounding environment; 0.9 wt.% sodium chloride (NaCl) aqueous solution was chosen as the electrolyte solution. Before EIS testing, the samples were immersed in the electrolyte for 1 h, 12 h, and 24 h, respectively. The test was measured at the open circuit potential (OCP) with an amplitude of 10 mV, and the scan frequency ranged from 100 kHz down to 0.01 Hz.

Immersion tests were performed to observe the corrosion morphology. The samples were immersed in 0.9 wt.% NaCl solution for different times. The optical images after different immersion times were recorded using an SZ680 stereomicroscope. The morphology of the matrix was obtained using a VHX-5000 super-high-magnification lens zoom 3D microscope (Keyence, Osaka, Japan) and a TESCAN MIRA2 SEM. Before observation, the corrosion products were removed by immersing the samples in a chromium trioxide solution (200 g/L) for 15 min at room temperature. The morphology of corrosion products was also characterized using the TESCAN MIRA2 SEM. Samples with corrosion products were firstly cold-mounted using epoxy resin and then mechanically polished to observe the cross-sections of the corroded areas. Before observation, a thin Au film was sputtered on the sample surface to avoid charging effects under SEM.

## 3. Results

### 3.1. Microstructure

The microstructures of the HOMO and LPSO samples are shown in Figure 1. In both samples, coarse gray phases and fine cubic phases can be detected on the grain boundaries. Compared with the HOMO samples, thin lamellar phases are found to precipitate in the grain interior of the LPSO samples. The XRD results and previous works [37,39,43,44] indicate that both the coarse phases and thin lamellar ones are the LPSO phases while the cubic phases are the RE-rich phases. Figure 1f lists the volume fraction of the coarse LPSO phases in the HOMO and LPSO samples calculated using Image-Pro software 6.0, and the values are roughly the same for both samples.

### 3.2. EIS Test Results

Figure 2 presents the Nyquist plots of the HOMO and LPSO samples immersed in 0.9 wt.% NaCl solution. Figure 2a,b presents the Nyquist diagrams of the HOMO samples. The EIS diagram of the HOMO sample after 1 h of immersion consists of a capacitive loop at high and medium frequencies and an inductive loop at low frequencies. After immersion for 12 and 24 h, both Nyquist diagrams are characterized by two capacitive loops at high and medium frequencies followed by two well-defined inductive loops at low frequencies (2.16–0.06 Hz and 0.06–0.01 Hz, respectively). Figure 2c,d presents the Nyquist plots of the LPSO samples. After immersion for 1 h, the shape of the Nyquist plots is similar to that of the HOMO samples, which also consist of one capacitive loop and one inductive loop. After immersion for 12 and 24 h, the shape of the Nyquist plots is different from that of the HOMO samples. At high and medium frequencies, there are also two capacitive loops. However, at low frequencies, there is an extra real part shrinkage (as indicated by the red box in Figure 2d) at 1.71–0.67 Hz and only one well-defined inductive loop at 0.67–0.01 Hz. In addition, only the existence of inductive impedance can lead to real part shrinkage with decreasing frequency [45]. Therefore, the real part shrinkage at 1.71–0.67 Hz suggests the presence of another inductive loop. Although the shape of the inductive impedance is different from the HOMO samples, the Nyquist plots of the LPSO samples after immersion for 12 and 24 h still contain two capacitive loops and two inductive loops.

### 3.3. Corrosion Morphology Observation

The abnormal shape of the inductive impedance in LPSO samples might be related to the corrosion morphology. Therefore, the corrosion morphology of the HOMO and LPSO samples were imaged and are presented in Figure 3, Figure 4, Figure 5, Figure 6, Figure 7, Figure 8 and Figure 9.

Figure 3 presents the optical images of the HOMO and LPSO samples after immersion for different durations. After immersion for 1 h, the surfaces of both samples are uniform and covered by gray corrosion products. After immersion for 12 and 24 h, the samples suffered from localized corrosion, leading to appreciable black areas on the sample surface. In addition, the filiform-like corrosion trace of black areas means that both samples suffered from filiform corrosion. Figure 3b depicts the fraction of black areas as a function of immersion times, which indicates that the black areas on the LPSO samples are larger than those on the HOMO samples.

Figure 4 presents the corrosion morphology of the HOMO and LPSO samples after 1 h immersion. In the HOMO and LPSO samples, shallow corrosion pits are detected, and there are more corrosion pits in the LPSO samples. Under higher magnification, it can be observed that these corrosion pits are adjacent to the LPSO phases. What is more, in the LPSO samples, the corrosion pits can be detected adjacent to both the coarse and thin lamellar LPSO phases. These results indicate that both the coarse and thin lamellar LPSO phases induced microgalvanic corrosion, and the LPSO samples suffered more severe microgalvanic corrosion.

Figure 5 displays the corrosion morphology of the gray areas after immersion for 24 h. The Mg matrix is lightly corroded in both the HOMO and LPSO samples, and only several shallow corrosion pits occur on the surface. This result indicates that the gray areas are well protected by the surface film.

Figure 6 and Figure 7 show the corrosion morphology of the black areas after immersion for 24 h. The 3D images reveal that the black areas are nonuniform (Figure 6a and Figure 7a). The lightly and severely corroded areas exhibit different corrosion traces. On the lightly corroded areas, shallow corrosion grooves can be observed, and the bottom of the groove is smooth (Figure 6b and Figure 7b), which is familiar with the corrosion trace of filiform corrosion [46]. On the severely corroded areas, a large deep corrosion pit can be observed, and there are many micropores on the bottom (Figure 6c and Figure 7c), which is familiar with the corrosion trace of pitting corrosion [12]. The nonuniform corrosion morphology indicates that both samples suffered from not only filiform corrosion but also pitting corrosion. Compared to the gray areas, the Mg matrix in the black areas is severely corroded. This result means that the surface film on the black areas is seriously damaged and cannot provide effective protection.

Figure 8 presents the cross-section of the black areas with corrosion products on both the HOMO and LPSO samples after immersion for 24 h. The corrosion products with different thicknesses were deposited on the Mg matrix of the black areas. The corrosion product layers are about a dozen microns thick on the lightly corroded areas while on the severely corroded areas, the thickness is over 100 μm. The XRD results (Figure 9) indicate that the corrosion products consist primarily of magnesium hydroxide (Mg(OH)_2_).

### 3.4. Equivalent Circuit and Fitting Results

As shown in Figure 3, Figure 4, Figure 5, Figure 6, Figure 7, Figure 8 and Figure 9, the corrosion morphologies of the HOMO and LPSO samples are similar. After immersion for 1 h, in both samples, small corrosion pits caused by microgalvanic corrosion can be observed adjacent to the LPSO phases. After immersion for 12 and 24 h, black areas appeared and exhibited nonuniform corrosion morphology.

Since the HOMO and LPSO samples exhibited similar corrosion morphology, the corrosion processes in the two samples were also similar, and the EIS results can be fitted by the same equivalent circuit. Our past study [42] has proved that these EIS results can be fitted by equivalent circuits, as shown in Figure 10. Figure 10a is the equivalent circuit used to fit the EIS results in 1 h, and each component has the following physical meaning: *R_s_* is the solution resistance, *R_t_* is the charge transfer resistance, *C_dl_* is the capacitance of the double electric layer, *R_L_* is the resistance of the ${\mathrm{Mg}}_{\mathrm{ads}}^{+}$ ion relaxation process, and *L* is the inductance of the ${\mathrm{Mg}}_{\mathrm{ads}}^{+}$ relaxation process. Figure 10b is the equivalent circuit used to fit the EIS results for 12 and 24 h, which contained dual inductive loops at low frequencies. Each component has the following physical meaning: *R_s_* is the solution resistance, *R_p_* is the pore solution resistance in the corrosion product film, *C_f_* is the capacitance of the corrosion product film, *R_t_* is the charge transfer resistance, *C_dl_* is the capacitance of the double electric layer, *R*_1_ and *L*_1_ are the resistance and inductance of the corrosion products and the ${\mathrm{Mg}}_{\mathrm{ads}}^{+}$ relaxation process on the severely corroded area, respectively, and *R*_2_ and *L*_2_ are the resistance and inductance of the corrosion products and the ${\mathrm{Mg}}_{\mathrm{ads}}^{+}$ relaxation process on the lightly corroded area, respectively. The fitting results are listed in Table 1 and Figure 1.

## 4. Discussion

### 4.1. Relationship between Inductive Impedances and Capacitance

Both the HOMO and LPSO samples with nonuniform corrosion morphology exhibited EIS with dual inductive loops. However, for the LPSO samples after longtime immersion (12 and 24 h), the inductive loop at 1.71–0.67 Hz appeared in the first quadrant rather than the fourth quadrant. The shape of the EIS diagram depends on the corrosion process of the Mg alloys, which contains several relaxation processes. Each relaxation process might be affected by other relaxation processes, and the influence of other relaxation processes can be analyzed through the fitting parameters. The EIS data with dual inductive loops were analyzed with the equivalent circuit, as shown in Figure 11a. Since the pore resistance *R_p_* is much lower than charge transfer resistance *R_t_* and the *R*_2_*L*_2_ branch has little effect on the impedance of the whole circuit under the higher frequencies (>0.67 Hz), the equivalent circuit can be simplified to the equivalent circuit, as shown in Figure 11b. Therein, *R* is approximately equal to the charge transfer resistance *R_t_*, *C* is equal to the capacitance of the surface film *C_f_* in parallel to the double electric layer *C_dl_*, and *R*_1_ and *L*_1_ are the resistance and inductance of the corrosion products and the ${\mathrm{Mg}}_{\mathrm{ads}}^{+}$ relaxation process on severely corroded areas. The impedance of the equivalent circuit Z can be expressed as follows:(1)$$Z=Z^{\prime }+{\mathrm{jZ}}^{\prime \prime }=\frac{a}{a^{2}{+b}^{2}}+j\frac{-b}{a^{2}{+b}^{2}}$$
with
(2)$$a=\frac{1}{R}+\frac{R_{1}}{R_{1}^{2}{+\omega }^{2}L_{1}^{2}}$$
(3)$$b=\omega C\;-\;\frac{{\omega L}_{1}}{R_{1\;}^{2}{+\omega }^{2}L_{1}^{2}}$$
where $Z^{\prime }$ represents the real part of *Z*, and $Z^{\prime \prime }$ represents the imaginary part of *Z*.

According to Equations (1)–(3), the imaginary part $Z^{\prime \prime }$ is affected by the value of *C*. Figure 11c,d shows the simulation results with variant parameters of *C*. In the case of a low value of *C* (1 × 10^−5^ F cm^−2^ and 3 × 10^−5^ F cm^−2^), the $Z^{\prime \prime }$ of the inductive loops is greater than zero at a low frequency, and the inductive loop can be observed in the fourth quadrant of the Nyquist plots. In the case of a high value of *C* (5 × 10^−5^ F cm^−2^ and 7 × 10^−5^ F cm^−2^), the $Z^{\prime \prime }$ of the inductive loop is less than zero, and the inductive loop can be observed in the first quadrant rather than the fourth quadrant. As listed in Table 2, the value of *C* in LPSO samples is greater than the HOMO samples, making $Z^{\prime \prime }$ constantly negative. As a result, the inductive loops are observed in the first quadrant for the LPSO samples.

### 4.2. Influence of Intragranular LPSO on Capacitance

As mentioned in Section 4.1, greater values of *C_f_* and *C_dl_* in the LPSO samples result in an abnormal shape of the inductive loops. Previous literature indicated that the values of *C_f_* and *C_dl_* are related to the condition of the surface film [25,47,48,49]. The interface of the Mg alloys can be schematically presented in Figure 12 [50,51,52]. The surface film consists of a thin inner film and a relatively thick outer film. The inner film mainly consists of magnesium oxide (MgO); this film can provide effective corrosion protection. The outer film mainly consists of Mg(OH)_2_; this film is porous and cannot provide effective corrosion protection. In the filmed areas, the compact MgO film effectively protected the Mg matrix from the corrosion of the NaCl solution. In the film-free areas, the Mg atom is oxidated to the Mg^2+^ ion and bonds with the hydroxide (OH^−^) to form Mg(OH)_2_ [15,53]:(4)$$\mathrm{Mg}\;\to {\;\mathrm{Mg}}^{2+}{+2e}^{-}\left(\mathrm{anodic}\;\mathrm{reaction}\right)$$
(5)$${2H}_{2}{O+2e}^{-}\to {2\mathrm{OH}}^{-}{+H}_{2}\left(\mathrm{cathodic}\;\mathrm{reaction}\right)$$
(6)$${2\mathrm{Mg}+2H}_{2}O\;\to {\;2\mathrm{Mg}(\mathrm{OH})}_{2}{+H}_{2}\left(\mathrm{overall}\;\mathrm{reaction}\right)$$

In this case, the capacitance of the double electric layer *C_dl_* and the surface film *C_f_* should also be related to the fraction of film-free areas *γ*. The capacitance of the double electric layer *C_dl_* can be calculated with the following equation:(7)$$C_{\mathrm{dl}}{=\gamma C}_{\mathrm{dl}}^{0}$$
where $C_{\mathrm{dl}}^{0}$ is the double electric layer capacitance per unit area.

The capacitance of the surface film *C_f_* can be calculated by:(8)$$C_{f}=\frac{\varepsilon_{0}\varepsilon }{l}$$
where $\varepsilon_{0}$ is the permittivity of the vacuum. ε is the permittivity of the surface film. *l* is the thickness of surface film areas. With the increase of film-free areas, more electrolyte permeates into the surface film, and the permittivity of electrolyte (≈80) is much greater than the permittivity of MgO and Mg(OH)_2_ (9.7 and 8.5, respectively). Meanwhile, the permittivity of the surface film ε also increases. Thus, the capacitance of the surface film *C_f_* is positively related to *γ*:(9)$$C_{f}=\frac{\varepsilon_{0}\varepsilon }{l}\propto \gamma$$

Equations (7) and (9) indicate that the values of *C_f_* and *C_dl_* are positively related to the fraction of film-free areas *γ*.

In addition, the charge transfer resistance *R_t_* can be calculated with the following equation:(10)$$R_{t}=\frac{R_{t0}}{\gamma }$$
where *R_t0_* is the charge transfer resistance per unit area. *γ* is negatively related to *R_t_*. According to the EIS fitting results (Table 1), the value of *R_t_* in the LPSO samples is lower than that in the HOMO samples, which revealed that the existence of intragranular LPSO created more film-free areas in the LPSO phases. This result is also confirmed by the observation of corrosion morphology. Compared to the gray areas, the surface film on the black areas has been severely damaged, and the Mg matrixes suffered from more severe corrosion (Figure 5, Figure 6 and Figure 7). From the corrosion morphology, one can deduce that the film-free areas in black areas are much larger than those in gray areas; the value of *γ* increases with the propagation of the black areas. The optical images in Figure 3 show that the black areas in the LPSO samples propagated much faster than that in the HOMO samples, which means that there are larger film-free areas in the LPSO samples than that in the HOMO samples. Figure 13 schematically shows the mechanism that the intragranular LPSO phases affected the surface film. During the initial time, both samples suffered from microgalvanic corrosion. The LPSO phases and the surrounding Mg matrix formed microcorrosion cells, and the Mg matrix acted as galvanic anodes. The microgalvanic corrosion destroyed the protective inner film on the anodic sites. The destruction of the inner film made the potential of corroded areas more negative than the adjacent areas, which formed micro-corrosion cells. Driven by the microcorrosion cells, the anodic sites propagated on the surface or to the depth direction. As a result, after longtime immersion, the microgalvanic corrosion developed into more severe localized corrosion (filiform corrosion and pitting corrosion). The corroded areas propagated to the adjacent areas, further destroying the protective inner film [12]. In the LPSO samples, the extra intragranular LPSO phases offered more galvanic cathodes for microgalvanic corrosion, which promoted the destruction of the protective inner film. Therefore, there are larger film-free areas in LPSO samples than that in HOMO samples, and the values of *C_f_* and *C_dl_* in the LPSO samples are also greater.

## 5. Conclusions

(1) After longtime immersion, the Nyquist plots of the HOMO and LPSO samples contained dual inductive loops at a low frequency. Both the HOMO and LPSO samples suffered from severe localized corrosion and exhibited nonuniform corrosion morphology. This result indicates that the occurrence of dual inductive loops should be related to the nonuniform corrosion morphology of VW64M Mg alloys.

(2) In the LPSO samples, the inductive loops at 1.71–0.67 Hz appeared in the first quadrant rather than the fourth quadrant. The analysis of the fitting parameters indicated that this abnormal shape of the inductive loops is related to greater values of the double-layer electric layer capacitance *C_dl_* and the surface capacitance *C_f_* in the LPSO samples.

(3) The existence of intragranular LPSO phases results in more severe microgalvanic corrosion and larger film-free areas, which lead to greater values of *C_dl_* and *C_f_* in the LPSO samples compared to the HOMO samples.

## Figures and Tables

**Figure 1 materials-16-00640-f001:**
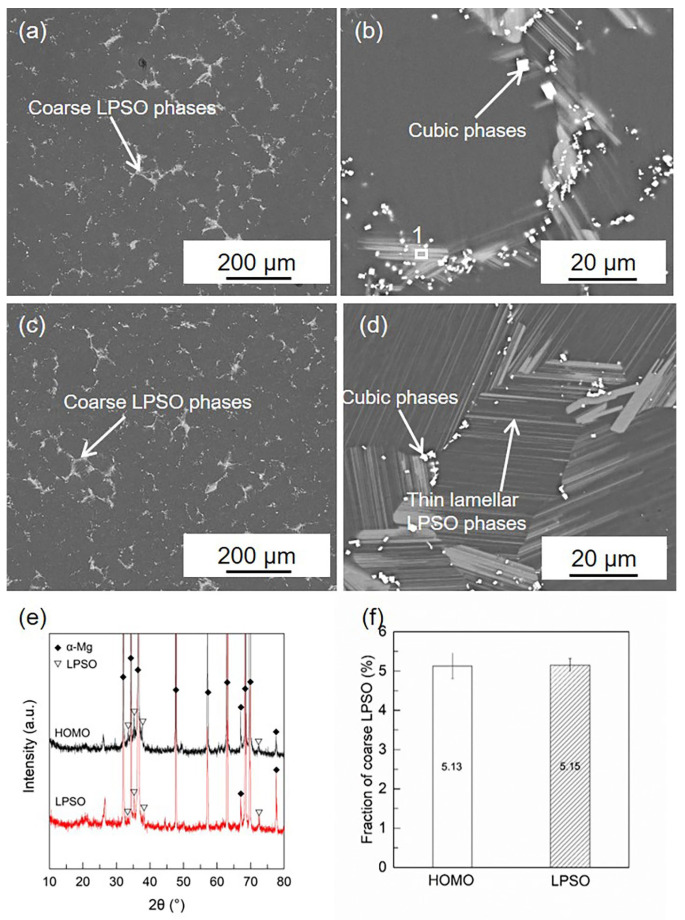
SEM images and XRD patterns of HOMO and LPSO samples: (**a**,**b**) SEM images of HOMO, (**c**,**d**) SEM images of LPSO samples, (**e**) XRD patterns of HOMO and LPSO samples, and (**f**) volume fraction of coarse LPSO phases.

**Figure 2 materials-16-00640-f002:**
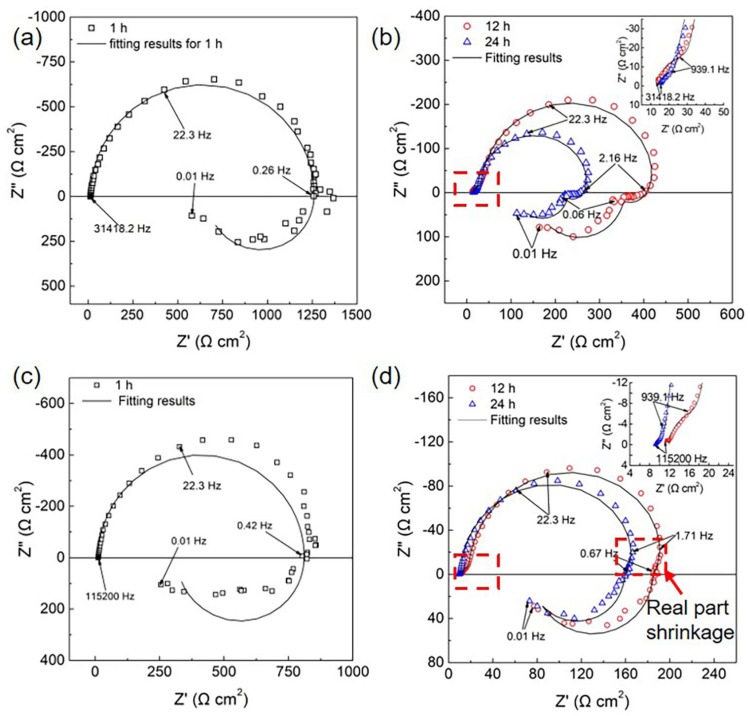
Nyquist plots in 0.9 wt.% NaCl solution: (**a**,**b**) the HOMO samples and (**c**,**d**) the LPSO samples. In (**b**,**d**), the enlargement of high-frequency parts was presented at the top left corner.

**Figure 3 materials-16-00640-f003:**
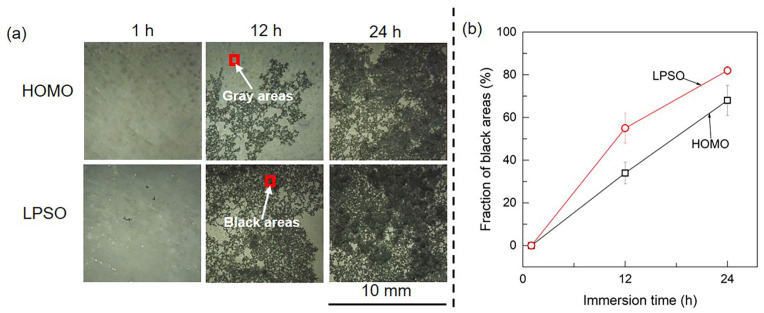
Optical images of the HOMO and LPSO samples: (**a**) optical images and (**b**) the fraction of black areas as a function of immersion time.

**Figure 4 materials-16-00640-f004:**
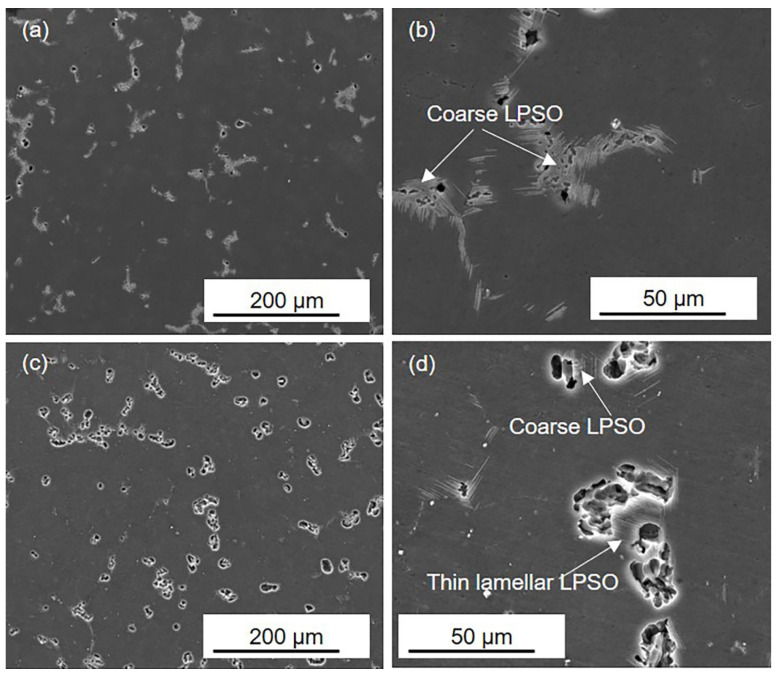
SEM images of (**a**,**b**) the HOMO and (**c**,**d**) LPSO samples after 1 h immersion.

**Figure 5 materials-16-00640-f005:**
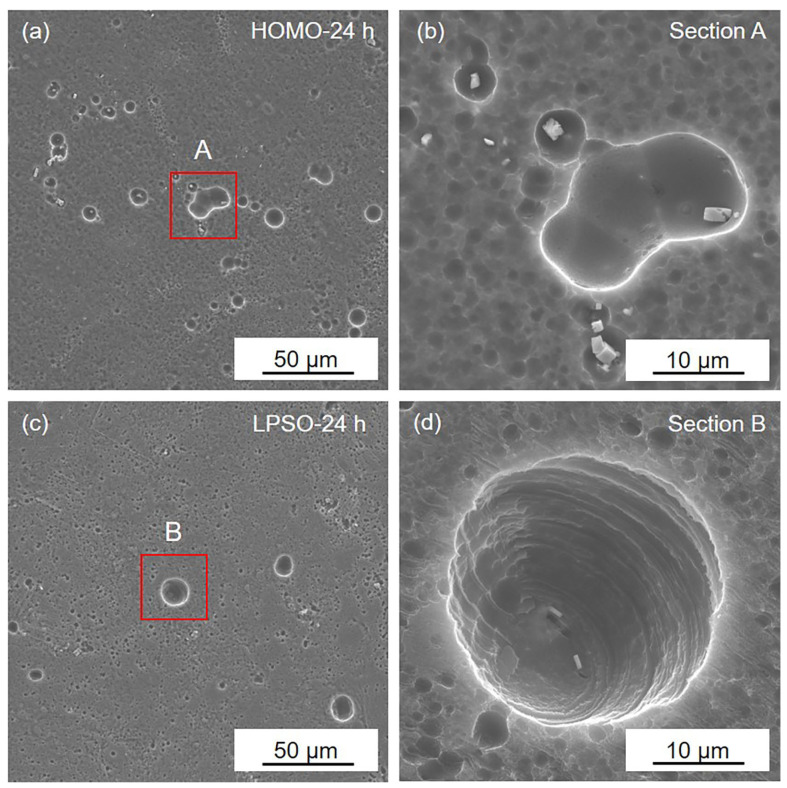
SEM images of the gray areas on the (**a**,**b**) HOMO and (**c**,**d**) LPSO samples after 24 h immersion.

**Figure 6 materials-16-00640-f006:**
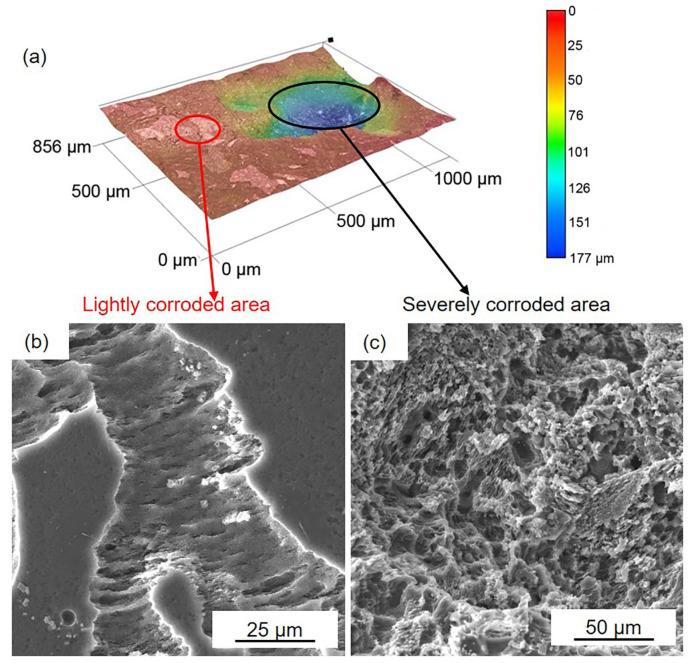
D optical and SEM images of the black areas on the HOMO samples after 24 h immersion: (**a**) 3D optical images, (**b**) SEM images in lightly corroded area, and (**c**) SEM images in the severely corroded area.

**Figure 7 materials-16-00640-f007:**
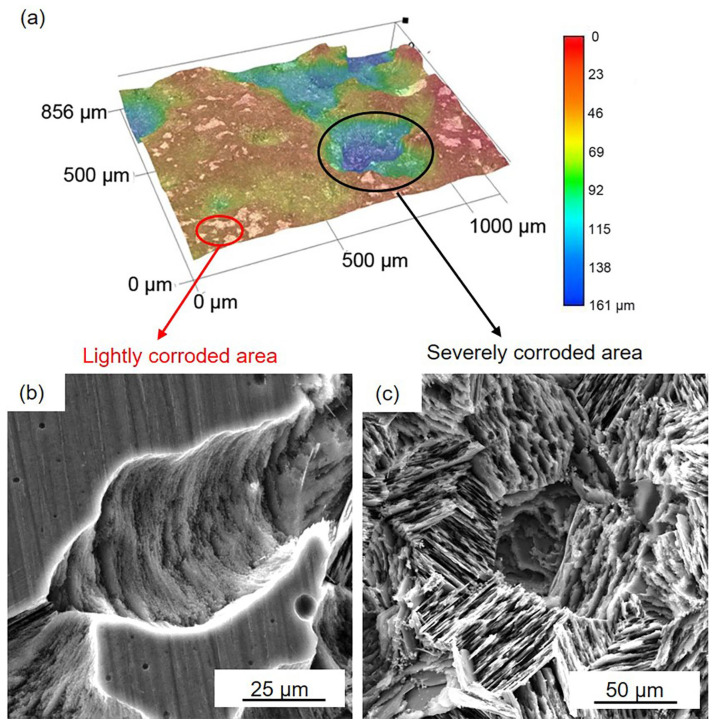
D optical and SEM images of the black areas on the LPSO samples after 24 h immersion: (**a**) 3D optical images, (**b**) SEM images in lightly corroded area, and (**c**) SEM images in the severely corroded area.

**Figure 8 materials-16-00640-f008:**
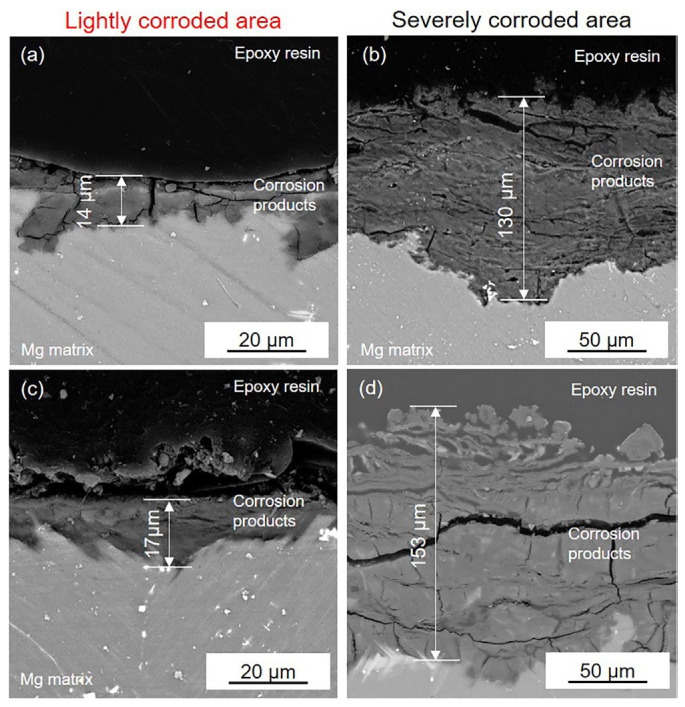
SEM images of the cross-section of (**a**,**b**) the black area on the HOMO samples and (**c**,**d**) the black area on the LPSO samples after 24 h immersion.

**Figure 9 materials-16-00640-f009:**
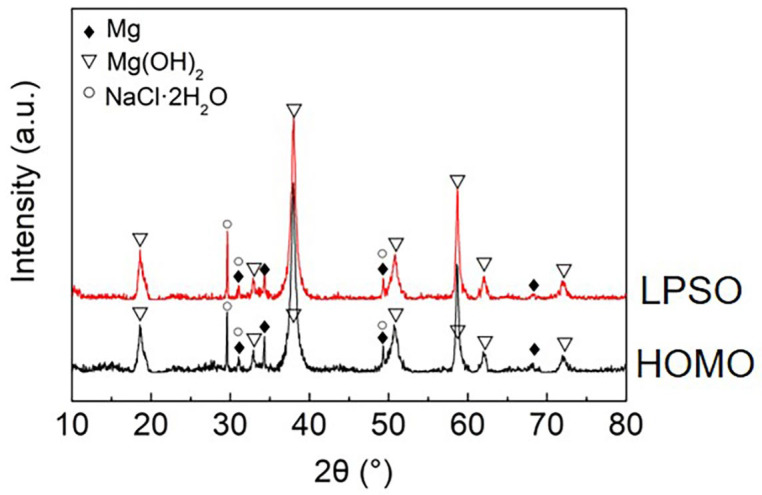
XRD patterns of the HOMO and LPSO samples after 24 h immersion.

**Figure 10 materials-16-00640-f010:**
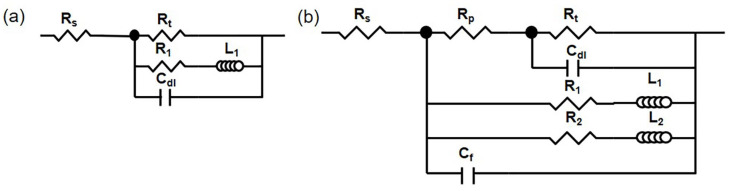
Equivalent circuit: (**a**) 1 h, (**b**) 12, and 24 h.

**Figure 11 materials-16-00640-f011:**
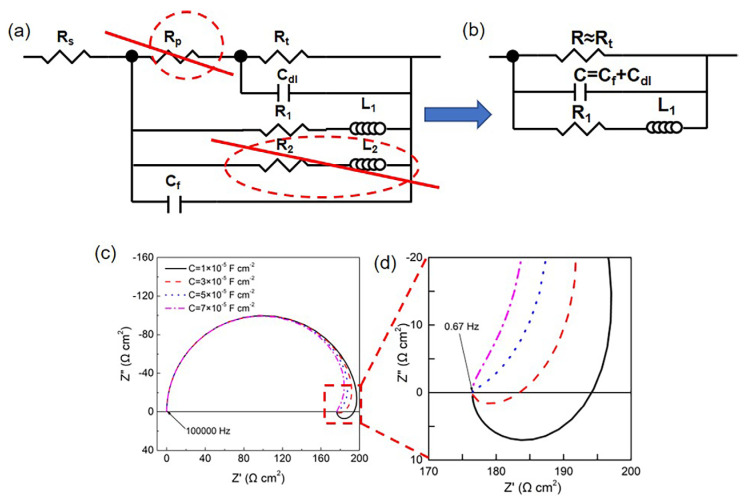
Influence of capacitance value on the inductive loops: (**a**) equivalent circuit for sample after immersion for 12 and 24 h; (**b**) equivalent circuit after simplified; and (**c**,**d**) simulation results for different C values, therein R = 200 Ω cm^2^, $R_{L}$ = 1500 Ω cm^2^, L = 100 H cm^2^, and frequency: 100,000–0.67 Hz.

**Figure 12 materials-16-00640-f012:**
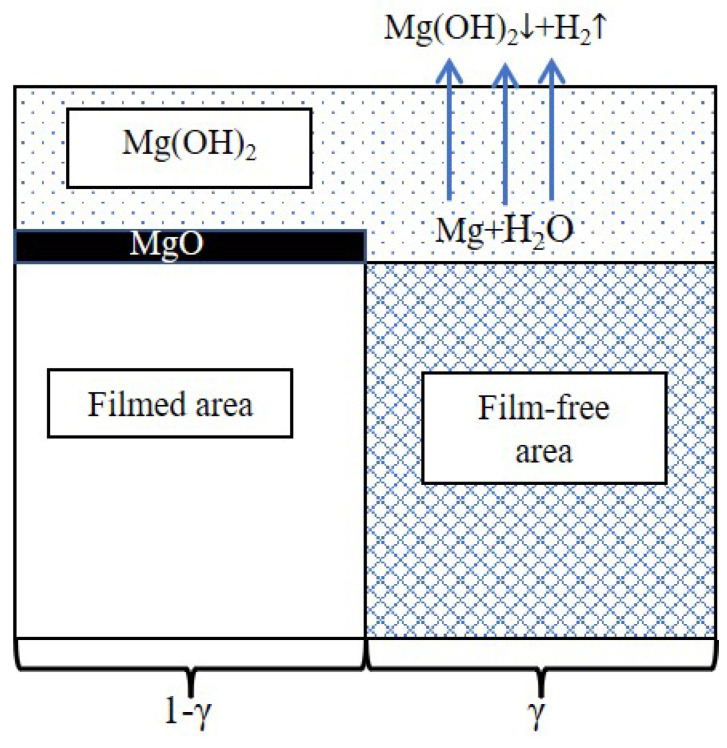
Schematic diagram of the surface film.

**Figure 13 materials-16-00640-f013:**
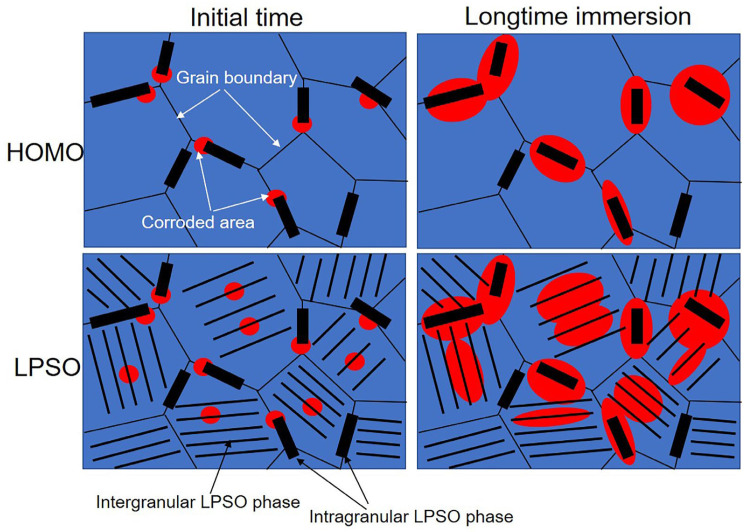
Schematic illustration shows intragranular LPSO phases’ influence on the surface film.

**Table 1 materials-16-00640-t001:** Fitting results of the EIS.

	HOMO-1 h	HOMO-12 h	HOMO-24 h	LPSO-1 h	LPSO-12 h	LPSO-24 h
$R_{s}$ (Ω cm^2^)	13	14	15	12	12	10
$C_{f}\;$(F cm^−2^)	-	5.06 × 10^−6^	9.30 × 10^−6^	-	1.43 × 10^−5^	2.66 × 10^−5^
$R_{p}\;$(Ω cm^2^)	-	29	14	-	13	6
$C_{\mathrm{dl}}\;$(F cm^−2^)	8.37 × 10^−6^	1.72 × 10^−5^	2.92 × 10^−5^	9.69 × 10^−6^	3.52 × 10^−5^	3.83 × 10^−5^
$R_{t}\;$(Ω cm^2^)	1247	395	254	800	179	159
$L_{1}\;$(H cm^2^)	12,227	167	108	4071	78	99
$R_{1}\;$(Ω cm^2^)	1357	1717	1016	494	1716	1576
$L_{2}\;$(H cm^2^)	-	4270	3153	-	1287	1626
$R_{2}\;$(Ω cm^2^)	-	232	177	-	103	113
*χ* ^2^	0.018	0.011	0.011	0.033	0.007	0.011

**Table 2 materials-16-00640-t002:** Fitting results of the *C* values.

	*C_f_* (F cm^−2^)	$C_{\mathrm{dl}}\;$	*C* (F cm^−2^)
HOMO-12 h	5.06 × 10^−6^	1.72 × 10^−5^	2.23 × 10^−5^
HOMO-24 h	9.30 × 10^−6^	2.92 × 10^−5^	3.85 × 10^−5^
LPSO-12 h	1.43 × 10^−5^	3.52 × 10^−5^	4.95 × 10^−5^
LPSO-24 h	2.66 × 10^−5^	3.83 × 10^−5^	6.49 × 10^−5^

## Data Availability

Not applicable.

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
