# Peer review of "Influence of Long-Period Stacked Ordered Phases on Inductive Impedance of Mg-Gd-Y-Zn-Zr-Ag Alloys"

_materials, 2023, doi:10.3390/ma16020640_

Round 1

Reviewer 1 Report

The article needs to be corrected in accordance with the indicated comments:

  1. From the text of the article it is not clear why it is necessary to conduct these studies. Need for research?

2. At the end of the paragraph "Introduction" indicate the purpose of the work and the objectives of the study.

3. What types of heat treatments have been subjected to samples of Mg-Gd-Y-Zn-Zr-Ag alloy. What are the heat treatment temperatures for alloys?

4. Increase the quality of drawings: resolution and font.

5. How the compositions of aggressive media were chosen during corrosion tests.

6. Influence of heat treatment regimes on the structure and corrosion properties of the alloy.

7. Provide a scientific justification for the effect of corrosion testing regimes on the corrosion resistance of alloys.

8. Influence of heat treatment temperatures on the corrosion properties of the alloy.

Reviewer 2 Report

The manuscript presents the interesting results concenring, the influence of long-period stacked ordered  phases on the electrochemical behaviour of an Mg-Gd-Y-Zn-Zr-Ag alloy in 0.9 wt.% NaCl. The experimental techniques are used correctly. The obtained results have been in detailed  analysed. In order to improve the manuscript the following comments should be explained:

1) In Fig.3 (a) the morphology of the Mg alloys after corrosion tests have been revealed. After 12 hours the filiform corrosion is clearly visible. It should be explain how is the mechanism of this corrosion.

2) The chemical reaction which caused the formation of the corrosion products should be written.

Round 2

Reviewer 1 Report

Dear colleagues!

There are no global comments on the text of the article. But, nevertheless, it is interesting how the authors selected the heat treatment modes of 510 °C for 24 h, etc.

Sincerely